



# Hyperspectral imaging sediment core scanning tracks high-resolution Holocene variations in (an)oxygenic phototrophic communities at Lake Cadagno, Swiss Alps

Paul D. Zander[1,2], Stefanie B. Wirth[3*], Adrian Gilli[4], Sandro Peduzzi[5] and Martin Grosjean[1]

[1]Institute of Geography and Oeschger Centre for Climate Change Research, University of Bern, Bern, 3012, Switzerland
[2]Climate Geochemistry Department, Max Planck Institute for Chemistry, Mainz, 55128, Germany
[3]Centre for Hydrogeology and Geothermics, University of Neuchâtel, Neuchâtel, 2000, Switzerland
[*]Present address: GEOTEST Ltd, 3052 Zollikofen, Switzerland
[4]Geological Institute, ETH Zurich, Zurich, 8092, Switzerland
[5]Department F.-A. Forel for environmental and aquatic sciences University of Geneva & Alpine Biology Centre Foundation, Bellinzona, 6500, Switzerland

*Correspondence to*: Paul D. Zander (paul.zander@mpic.de)

## Abstract

Pigments produced by anoxygenic phototrophic bacteria are valuable proxies of past anoxia in lacustrine and marine environments. Pigment measurement typically requires time-consuming and costly chemical extractions and chromatographic analyses, which limits the temporal resolution of paleoenvironmental reconstructions based on sedimentary pigments. Here, we evaluate the potential of *in-situ* hyperspectral imaging (HSI) core scanning as a rapid, non-destructive method to document high-resolution changes in oxygenic and anoxygenic phototrophic communities at meromictic Lake Cadagno, Switzerland. Three distinct groups of pigments can be detected with the HSI method in the sediments of Lake Cadagno; each pigment group represents a different phototrophic community. Oxygenic phototrophs are indicated by total chloropigments (TChl; chlorophyll-*a, -b* and derivatives). Two types of anoxygenic phototrophs were distinguished - purple sulfur bacteria (PSB), represented by bacteriochlorophyll-*a*, and green sulfur bacteria (GSB), represented by bacteriochlorophylls-*c*, -*d*, and -*e*. HSI pigment indices were validated by pigment measurements performed on extracted samples using spectrophotometer and high-performance liquid chromatography (HPLC). Bacteriochlorophylls were present throughout the past 10 kyr, confirming geochemical evidence of nearly continuous stratification and sulfidic conditions at Lake Cadagno. Major shifts in the anoxygenic phototropic communities are recorded at decadal to millennial scales. GSB and PSB communities coexisted from 10.2-3.4 kyr BP, with dominance of PSB over GSB from 8.8-3.4 kyr BP indicating strongly stratified conditions in the lake and strong light radiation at the chemocline. From 3.4-1.3 kyr BP, PSB were mostly absent, and GSB became dominant, implying lower light intensity at the chemocline due to a combination of factors including deforestation in the lake surroundings, increased flood frequency, cooler climatic conditions, and changes in groundwater solute concentrations. The high-resolution HSI data show that frequent flood events and mass movements disturbed the chemocline and the anoxygenic bacterial communities, and that the PSB were particularly sensitive and slow to recover following these disturbance events.





This study demonstrates for the first time that HSI can detect GSB related pigments, making the method uniquely valuable as a rapid tool to study samples containing pigments of both oxygenic and anoxygenic phototrophs.

## 1 Introduction

Photosynthetic pigments preserved in sediments are a valuable source of information on past environmental conditions, and can be used to track past variations in multiple environmental parameters including trophic levels, primary productivity, light availability, organic matter preservation, and redox conditions (Leavitt and Hodgson, 2002). Pigments derived from anoxygenic phototrophic bacteria are particularly valuable as indicators of anoxic and sulfidic conditions within the photic zone of aquatic environments (Sinninghe Damsté and Schouten, 2006). Phototrophic sulfur bacteria live at, or below, the chemocline in stratified water bodies and typically use reduced sulfur as an electron donor during photosynthesis (Van Gemerden and Mas, 1995). There are two main groups of anoxygenic phototrophic bacteria: green sulfur bacteria (GSB; *Chlorobiaceae*) and purple sulfur bacteria (PSB; *Chromatiaceae* and *Ectothiorhodospiraceae*) (Sinninghe Damsté and Schouten, 2006). These bacteria utilize bacteriochlorophyll (BChl) pigments to photosynthesize and also produce distinct accessory carotenoid pigments. PSB produce BChl-*a* and -*b* and the diagnostic carotenoid okenone. In contrast, GSB produce BChl-*c, -d,* or -*e* and the carotenoids isorenieratene and chlorobactene. The light absorption of these pigments is strongly influenced by the ecological niche of these bacteria, that is deeper water with lower light intensity compared to oxygenic phototrophs in the near-surface (mixolimnion). Therefore, the phototrophic sulfur bacteria utilize pigments that absorb light in the maxima of the light spectrum in deeper water, and do not absorb light in the same range as algae and cyanobacteria living in the overlying oxygenated mixolimnion. Purple sulfur bacteria require higher levels of light and are tolerant of small amounts of oxygen, and therefore, PSB usually live above the GSB in the water column (Van Gemerden and Mas, 1995). The GSB, in contrast, are strict anaerobes and can grow in extremely light-limited environments (Repeta et al., 1989; Crowe et al., 2014). Pigments from these bacteria are particularly useful as paleo-redox proxies, because these biomarkers provide an unambiguous signal of anoxic conditions within the water column, whereas many other geochemical indicators of anoxia can also be diagenetically produced by anoxic conditions within shallow sediments. Furthermore, the different light requirements of the two groups make it possible to reconstruct changes in light availability at the chemocline or chemocline depth (Fox et al., 2022).

Hyperspectral imaging (HSI) core scanning has recently emerged as a rapid, non-destructive method to determine the relative abundance of bulk pigment groups in sediments based on their distinct spectral signatures in the visible and near infrared range of light (Butz et al., 2015). Compared to conventional techniques that rely on wet-chemical extractions and detailed chemical measurements, the HSI method is much more efficient in terms of time and cost, and it has the advantage of producing ultra-high-resolution data at sub-mm resolution, something that is not possible with other, wet-chemical methods. Studies have demonstrated that chlorophyll-*a* (and it's diagenetic products; hereafter referred to as TChl), as well as bacteriopheophytin-*a*



(a diagenetic product of BChl-*a*), can be measured with the HSI technique and validated by conventional high-performance liquid chromatography (HPLC) measurements (Butz et al., 2015, 2017; Zander et al., 2022). However, to date, it has not been shown if GSB-related pigments (i.e. BChl-*c, -d,* and -*e)* can also be detected using this technique. This would be a particularly useful methodological advance because BChl-*c, -d,* and -*e* cannot be distinguished from TChl using spectrophotometric techniques on pigment extracts (Picazo et al., 2011), meaning there is currently no rapid method for assessing the abundance

of TChl or BChl-*c*, -*d*, and -*e* in samples that contain a mixture of both pigment groups.

In this study, we investigate the sediments of Lake Cadagno, a meromictic lake in the Swiss Alps with a well-studied sedimentary record (Wirth et al., 2013; Berg et al., 2022). This site is unique for its sulfidic bottom waters and dense populations of diverse phototrophic sulfur bacteria at the chemocline (Tonolla et al., 2017, 2005; Peduzzi et al., 2012, 2011).

Therefore, it is an ideal site to test the feasibility of using HSI to measure GSB-related bacteriochlorophylls in sediments that contain a mixture of pigments from both oxic and anoxic phototrophs. Furthermore, we generate high-resolution records of pigment groups spanning the 12.5-kyr-long sediment record in order to assess the factors that affected the relative abundances of different phototrophic communities on both short-term (event to decadal) and long-term (millennial) timescales.

## 2 Lake Cadagno and its sediment record

### 2.1 Setting and modern physicochemical and ecological properties

Lake Cadagno is a small (0.26 km$^2$) crenogenic meromictic lake located in the Piora Valley in the central Alps of Switzerland (46°33'01" N, 8°42'41" E) at an altitude of 1921 m asl (Fig. 1A). The 21-m-deep lake basin was liberated by the glacier ~13 kyr ago (Wirth et al., 2013). The catchment geology primarily consists of high- to ultrahigh-grade metamorphic rocks such as para- and orthogneisses, as well as amphibolites or hornblende schists (Bianconi et al., 2015). Particularly important for the

lake's extraordinary characteristics are, however, Triassic gypsum-bearing dolomite and graywacke, the so-called Piora Zone, that outcrops at the southern border of the lake. Subaquatic springs within the Triassic rocks that dip below the lake maintain a flow of solute-rich groundwaters to the lake. As a result, a strong density gradient exists in the water column leading to permanent stratification with an oxic mixolimnion, an anoxic monimolimnion and a chemocline at a depth of 10 to 14 m separating the two water masses (Del Don et al., 2001). Figure 1B shows water-column profiles of dissolved O$_2$, H$_2$S, as well

as the electrical conductivity and water temperature, demonstrating the contrasting chemical and physical properties of Lake Cadagno above and below the chemocline (Dahl et al., 2010). The oxic zone is fed primarily by surface waters flowing on the crystalline metamorphic bedrock, is poor in solute load, and oligo- to mesotrophic. Primary production in the oxic epilimnion is dominated by diatoms, chlorophytes and cyanobacteria (Bossard et al., 2001; Tonolla et al., 2017; Camacho et al., 2001). Sediment-trap data showed that only about one third of this algal biomass passes through the chemocline to a depth of 13.5 m

(Schanz and Stalder, 1998). The chemocline hosts dense populations of phototrophic sulfur bacteria of the families *Chromatiaceae* (purple sulfur bacteria; PSB) and *Chlorobiaceae* (green sulfur bacteria; GSB) (Danza et al., 2018). Despite



representing approximately 20% of the total lake volume, the chemocline is responsible for nearly half of the total carbon

photo assimilation within the lake, indicating that the phototrophic sulfur bacteria play a significant role in the ecosystem

functioning (Camacho et al., 2001). Investigations spanning the past three decades of the phototrophic bacteria populations in

the chemocline and monimolimnion detected only brown-colored GSB (no green-colored strains) using both 16S rRNA gene

clone library analysis and direct observations of lake samples (Tonolla et al., 2005; Halm et al., 2009; Gregersen et al., 2009;

Ravasi et al., 2012). Since the GSB observed in the recent decades are from brown-colored strains, notably *Chlorobium spp.*,

they all contain isorenieratene and are considered typical for low light availability (Danza et al., 2018; Posth et al., 2017). In

accordance with their low light tolerance, the GSB can also be found throughout the monimolimnion, though their energy

pathways may shift to include fermentation of polyglucose as an energy source when they sink below 14 m depth (Danza et

al., 2018; Habicht et al., 2011).

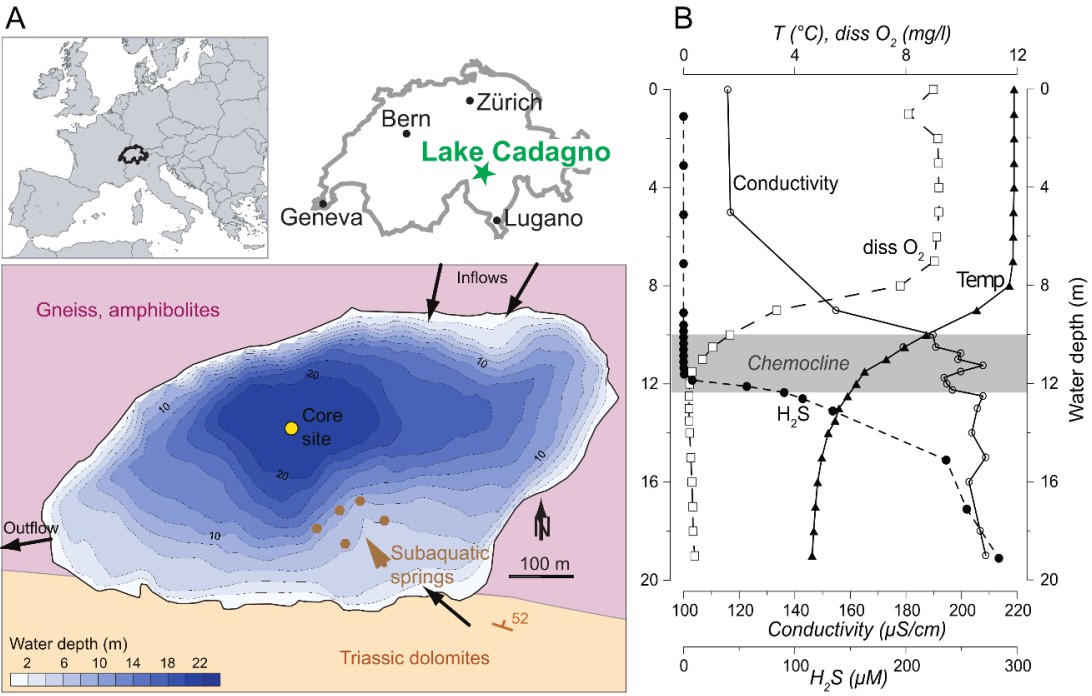

**Figure 1: A) Study site location within Europe and Switzerland, and bathymetric map of Lake Cadagno with simplified bedrock**
**geology and hydrogeological features depicted. B) Water-column physicochemical profiles highlighting the oxic mixolimnion, the**
**chemocline, and the sulfidic monimolimnion (Dahl et al., 2010, measured in Sep 2006).**

## 2.2 Sediment record and sediment properties

This section provides a short summary of sedimentological and stratigraphic work described in full and detail in Wirth et al.

(2013). The complete sedimentary record was retrieved for the first time in 2009 using a UWITEC piston coring system. To

guarantee a complete sediment sequence without gaps in between sections, twin cores were taken with a vertical offset of 1.5



m. In addition, several gravity short cores were taken to recover an undisturbed sediment surface. Nevertheless, sediments are disturbed in the top 5 cm of the short core that was used for this study, corresponding approximately to the interval from the time of coring (2009) to 1963 AD; based on $^{137}$Cs dating published in Wirth et al. (2013). The modern conditions in the lake are, therefore, not represented in the sedimentary data. The core chronology is based on nine radiocarbon ages as well as on

$^{137}$Cs dating for the most recent 60 years. The age-depth model was established using the regular sediments only (i.e., event deposits were removed) as event layers are deposited within hours to days (Sabatier et al., 2022). For age-depth modeling, a smoothed cubic spline interpolation between dating points using the Clam software was applied (Blaauw, 2010). We use the age-depth model published in Wirth et al, 2013 without modification.

XRF core scanning was carried out using an Avaatech core scanner with a rhodium tube as X-ray source. Vertical step resolution and slit opening was 1 mm, and the horizontal slit opening was 12 mm. Exposure time was 20 s, applied current 2000 µA, and applied voltage 30 kV. Semi-quantitative counts from XRF core scanning were calibrated to Mo concentrations measured via ICP-MS on 19 discrete samples.

The composite sediment record has a length of 10.5 m, and spans about the past 12.5 kyr (Wirth et al., 2013). The lowermost meter of recovered sediments is of gray color and purely clastic, representing the end of the deglaciation sequence. Between 9.5 and 9 m core depth, the sedimentary facies is characterized by a decreasing amount of clastic material and the organic content begins to increase. This section represents the redox-transition interval from oxic to sulfidic conditions, which occurred around 9.8 kyr BP (Wirth et al., 2013). From 9 m core depth to the top of the sequence, the sediments are characterized by the

alternation of three different lithologies: regular lacustrine sediments, flood deposits, and mass-movement deposits. Overall, only 2.3 m of the recovered sediment record consist of regular authigenic lacustrine sediments. This lithology is characterized by a high organic carbon content of 8 to 19 wt% and a dark brown to blackish color with little structure. Only at a few stratigraphic levels in the younger half of the sequence are laminations visible in intervals of a few centimeters. The majority of the sediment record consists of event deposits. These deposits are produced by floods as the result of heavy precipitation in

the lake's catchment or by subaquatic mass movements generated by the failure of the lake slopes due to earthquake-induced shaking or overloading (e.g. Schnellmann et al., 2002; Sabatier et al., 2022). Flood deposits represent sediments that are flushed into the lake. They are composed of allogenic mineral grains (primarily from gneiss) and are characterized by an only faintly expressed upwards grain-size grading. These deposits vary in thickness from about one mm to 13 cm. Mass-movement deposits (MMDs) represent sediments that were remobilized and redeposited within the lake and thus consist of a mix of regular

sediments and flood deposits. These deposits are up to 70 cm thick. The average sedimentation rate in the regular sediments is 0.19 mm/yr. However, rates are substantially lower in the older part (e.g. 0.07 mm/yr between 9 and 8 kyr BP) and become higher toward the top (e.g. 0.38 mm/yr between 2 and 1 kyr BP).





Based on the previous studies that investigated Lake Cadagno's sediment record, it is known that the lake has been meromictic
with a sulfidic monimolimnion throughout the Holocene. These conditions are documented by elevated sedimentary
molybdenum (Mo) concentrations of >26 ppm, Mo isotopic compositions characteristic for euxinic conditions, and the
presence of phototrophic sulfur bacteria pigments (isorenieratene and okenone) and DNA (Dahl and Wirth, 2017; Ravasi et
al., 2012; Wirth et al., 2013; Dahl et al., 2013).

## 3 Methodology

### 3.1 Hyperspectral Imaging core scanning

Hyperspectral imaging (HSI) core scanning was performed using a Specim PFD-CL-65-V10E linescan camera following the
procedure developed by Butz et al. (2015). The spatial resolution of the images is 70 μm pixel size, and the spectral resolution
is 1.6 nm. The resulting images were processed using ENVI (Exelisvis ENVI, Boulder, Colorado). Reflectance from 471-950
nm was normalized using a $BaSO_4$ white reference plate as a white reference and an image taken with a closed camera shutter
as a dark reference. Data were removed from the hyperspectral images based on user-selected reflectance thresholds to exclude
areas of the sediment core with cracks (pixels with less than 3.8% reflectance in the range 600-800 nm) or unusually reflective
minerals (pixels with greater than 28% reflectance between 600-800 nm).

Relative absorption band depth (RABD) indices were calculated to estimate the relative abundance of pigment groups based
on their diagnostic absorption troughs (Fig. 2). Spectral endmember analysis was used to identify troughs associated with three
types of photosynthetic pigments: Chl-*a* (absorption at 660-675 nm; Rein and Sirocko, 2002), BChl-*a* (absorption at 840-845
nm; Butz et al., 2015; Oren, 2011), and BChl-*c*, -*d*, and -*e* (absorption 710-745 nm; Hubas et al., 2011; Oren, 2011). Multiple
RABD formulas were tested to quantify pigment abundance, and were compared with high-precision liquid chromatography
measurements of pigment extracts from discrete samples to select the RABD formulas that effectively captured pigment
abundance (Fig. S1). Based on this analysis, the following formulas were used:

$$RABD_{670} = \left( \frac{19 \times R_{640} + 16 \times R_{695}}{35} \right) / R_{670} \tag{1}$$

$$RABD_{710-730max} = \max_{710 < i < 730} \frac{\left( \frac{X \times R_{695} + Y \times R_{765}}{X + Y} \right)}{R_i} \tag{2}$$


$$RABD_{842} = \left( \frac{33 \times R_{790} + 36 \times R_{900}}{99} \right) / R_{842} \tag{3}$$



Where $R_\lambda$ is the reflectance at the wavelength ($\lambda$); in eq. 2, X is the number of spectral bands between $R_{765}$ and the trough minimum ($R_i$), and Y is the number of spectral bands between the trough minimum and $R_{695}$.


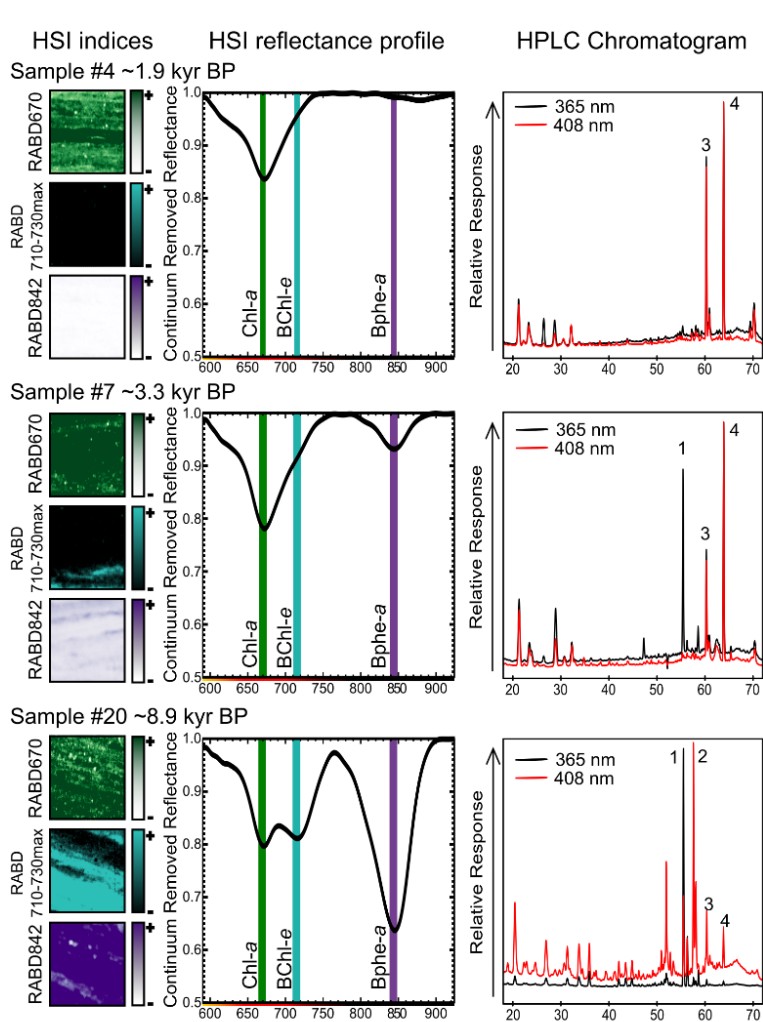

**Figure 2: Examples of sediment samples (1 cm², left) taken for pigment calibration comparing RABD index maps, the mean reflectance profile over the 1 cm² sample area (middle, shown as continuum removed reflectance), and HPLC chromatograms (right). Absorption maxima for Chl-*a* (green), BChl-*e* (light blue) and Bphe-*a* (purple) are labeled by vertical lines on the HSI**
**reflectance profiles. Peaks in HPLC chromatograms are labeled: 1) Bacteriopheophytin-*a*; 2) isorenieratene; 3) pheophytin-*a* (derivative of chlorophyll-*a*); 4) pyropheophytin-*a* (derivative of chlorophyll-*a*).**

RABD$_{670}$ represents the relative abundance of chlorophyll-*a, -b* and derivatives (TChl), and thus, is a proxy for primary production within the oxic mixolimnion of the lake. This index, or similar variations, has been widely used in investigations of lake sediments (Michelutti and Smol, 2016; Zander et al., 2022). RABD$_{710-730max}$ is reported here for the first time, and

represents the relative abundance of BChl-*c, -d,* and *-e*, as well as their derivatives. Therefore, it is a proxy for the abundance of GSB. RABD$_{842}$ represents the relative abundance of BChl-*a*, which is preserved in sediments as bacteriopheophytin-*a*



(Bphe-*a*). This index has been utilized as a proxy for PSB in several studies utilizing HSI on sediment cores (e.g. Butz et al., 2016; Makri et al., 2020).

## 3.2 Pigment measurements for calibration of HSI indices

Pigments were extracted from 22 samples taken from the sediment core from stratigraphic levels that covered as wide a range of the RABD values as possible. Each sample represents approximately 1 cm$^2$ on the core surface and contained approximately 0.2-0.6 g of dry sediments. Pigments were extracted with 100% Acetone using a modified version of the method described in Amann et al. (2014). A spectrophotometer (Shimadzu UV-1800) was used to measure Bphe-*a* in extracts by applying the extinction coefficient from Fiedor et al. (2002). The same extracts were also measured using an Agilent Infinity 1260 series

HPLC equipped with a G7117C Diode-Array Detection detector using the method of (Sanchini and Grosjean, 2020). We used standards to identify and quantify concentrations of chlorophyll-*a,* pheophytin-*a*, pyropheophorbide-*a*, pheophorbide-*a*, and pyropheophytin-*a*. All of these pigments were combined to estimate the concentration of TChl*,* which was used as the target of the RABD$_{670}$ calibration. Bphe-*a* concentrations were estimated using a calibration based on a BChl-*a* standard, which yields results that are highly correlated with the concentrations obtained from the spectrophotometer method (r=0.97, p<0.05),

but are overestimated due to the higher absorption coefficient of BChl-*a* compared to Bphe-*a* (Oelze, 1985). For this reason, we use the spectrophotometer data for calibration of the RABD$_{842}$ index. Additionally, the spectrophotometer data track all bacteriopheopigments that absorb in the same band (745 nm in Acetone), which is comparable to the *in-situ* measurements of the HSI method. Finally, the carotenoid isorenieratene was identified in the HPLC chromatograms based on its retention time and spectral profile (Itoh et al., 2003; Fuciman et al., 2010). This carotenoid is produced by brown-colored strains of GSB

(Sinninghe Damsté and Schouten, 2006; Hopmans et al., 2005), and thus serves as a proxy for their abundance. We report isorenieratene as a relative index (peak area per g dry sediment). We tested the correlation between isorenieratene and RABD$_{710\text{-}730max}$ to validate the new HSI index as a proxy for GSB. Unlike the other two HSI pigment indices, a quantitative concentration model for the RABD$_{710\text{-}730max}$ index was not calculated. This is because it was not possible to determine total BChl-*c*, -*d*, and -*e* concentrations due to lack of commercially available standards, numerous homologues and degradation

products, and the difficulty of distinguishing these GSB-derived bacteriochlorophylls from degradation products of chlorophyll-*a* or other chloropigments with similar UV-VIS absorbance profiles in solvent.

Calibration models and statistics were calculated in using R 4.2 (R Core Team, 2022). One sample, which was taken from the oldest clastic unit, contained no measurable pigments, and thus was excluded from the data analysis. Linear regression was

performed with data from the remaining 21 samples to calculate calibration models where the HSI RABD indices are the predictor and measured pigment concentrations as the response variables. The root mean square error of prediction (RMSEP) was estimated using a 10-fold cross validation method. In addition to our new pigment measurements, we compare our HSI indices for PSB and GSB to previously published data on DNA and carotenoid pigments for these two bacterial groups (Wirth et al., 2013; Ravasi et al., 2012). The published data were reported as relative ratios of each group. To compare our HSI data





with these published data, we calculated a 100-year moving average of the HSI data, and then the relative proportion of the
PSB and GSB HSI indices by standardizing the range of each index and then calculating the ratio of the two indices.

## 4 Results and discussion

### 4.1 Hyperspectral imaging as a tool to measure sedimentary pigments

The comparison between pigment measurements on extracted samples and the HSI RABD indices validates the use of the HSI
method to track abundances of the three pigment groups investigated here: TChl, Bphe-*a*, and BChl-*c*, -*d*, and -*e* (dataset -
Zander et al., 2023). The concentration of Bphe-*a* is highly correlated with RABD$_{842}$ (Fig. 3A), and the calibration model is
robust, with an estimated error of 10.4 %. This result is confirmed whether measurements from the spectrophotometer or HPLC
are used (r = 0.96 or r = 0.92, respectively; p < 0.001), which strengthens confidence in the interpretation. The concentration
of TChl is significantly correlated with RABD$_{670}$ (r = 0.81, p < 0.001), though the calibration model has more error compared
to the Bphe-*a* model (Fig. 3B). Isorenieratene is significantly correlated with the new index, RABD$_{710\text{-}730\text{max}}$ (Fig. 3C; r = 0.82,
p < 0.001), though the skewed distribution of concentrations increases uncertainty in interpreting whether or not the
relationship is linear. Isorenieratene is a proxy for GSB abundance (Sinninghe Damsté and Schouten, 2006) and is found in all
GSB strains in modern Lake Cadagno. Thus, the significant relationship with isorenieratene abundance validates the use of the
RABD$_{710\text{-}730\text{max}}$ index as a proxy for GSB. However, because we could not quantify total Bchl-*c*, -*d*, and -*e*, a quantitative
calibration of the RABD$_{710\text{-}730\text{max}}$ is not possible with our current dataset. The indirect relationship between isorenieratene
(proxy for GSB) and RABD$_{710\text{-}730\text{max}}$ shows more scatter compared to the other pigment/HSI comparisons, likely because the
absorption feature measured by RABD$_{710\text{-}730\text{max}}$ is attributed BChl-*c*, -*d*, and -*e* rather than isorenieratene. Thus, this relationship
could be affected by factors that change the ratio of sedimentary isorenieratene to BChl-*c*, -*d*, and -*e*, such as changes in GSB
communities or preservation conditions. Isorenieratene is specific to brown-colored GSB, which live in relatively dark
conditions (Manske et al., 2005). If the population shifted towards green-colored GSB in the past, then isorenieratene would
decrease (replaced by chlorobactene) and BChl-*c* would also become more abundant, with declining portions of Bchl-*e*.
Changes in the exact BChl pigments could affect the RABD$_{710\text{-}730\text{max}}$ index values because specific BChl pigments have
different extinction coefficients (Namsaraev, 2009). Our index does account (to some extent) for shifts in the location of peak
absorption wavelengths by using the flexible, maximized version of the RABD formula (eq. 2 in Section 3.1) that uses the
maximum trough depth from 710-730 nm. An additional factor that may weaken the relationship between the RABD$_{710\text{-}730\text{max}}$
index and isorenieratene is the fact that Chl-*a* pigments also absorb light in the bands utilized by the RABD$_{710\text{-}730\text{max}}$ formula.
We attempted to minimize this effect by testing different RABD formulas and comparing the formula variations with the
HPLC pigment data. In the HPLC data, isorenieratene and TChl are uncorrelated (r = 0.01), and we used this information to
select RABD formulas that also produced a low correlation between the GSB and TChl HSI indices (r = 0.04 for the 21 sample
locations, Fig. S1). The RABD formulas that use a trough endpoint located between the TChl and BChl-*c*, -*d*, and -*e* absorption





troughs (695 nm) show less correlation to one another, consistent with the HPLC pigment data, whereas formulas using a wider set of endpoints tended to show a positive correlation for the two troughs (Fig. S1).

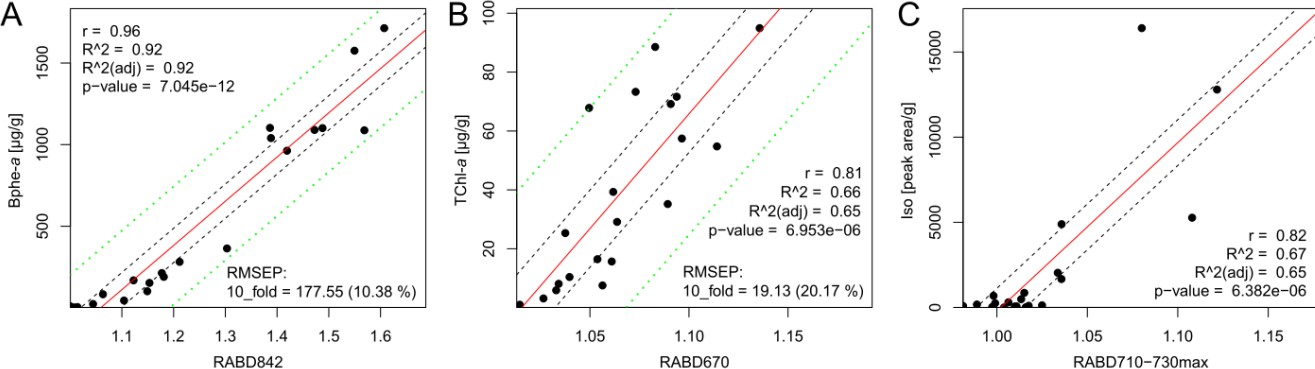

**Figure 3: Correlation of pigment measurements with HSI indices. Black dashed lines indicate the confidence interval of the**
**regression line, green dashed likes indicate the confidence interval of the calibration model predictions. No calibration model was used for RABD$_{710-730max}$ (panel C).**

The HSI indices for PSB and GSB show a similar pattern over the past 12 kyr when compared with the DNA and carotenoid pigments from Wirth et al. (2013), further validating the HSI indices RABD$_{710-730max}$ and RABD$_{842}$ as proxies for GSB and PSB, respectively (Fig. 4).


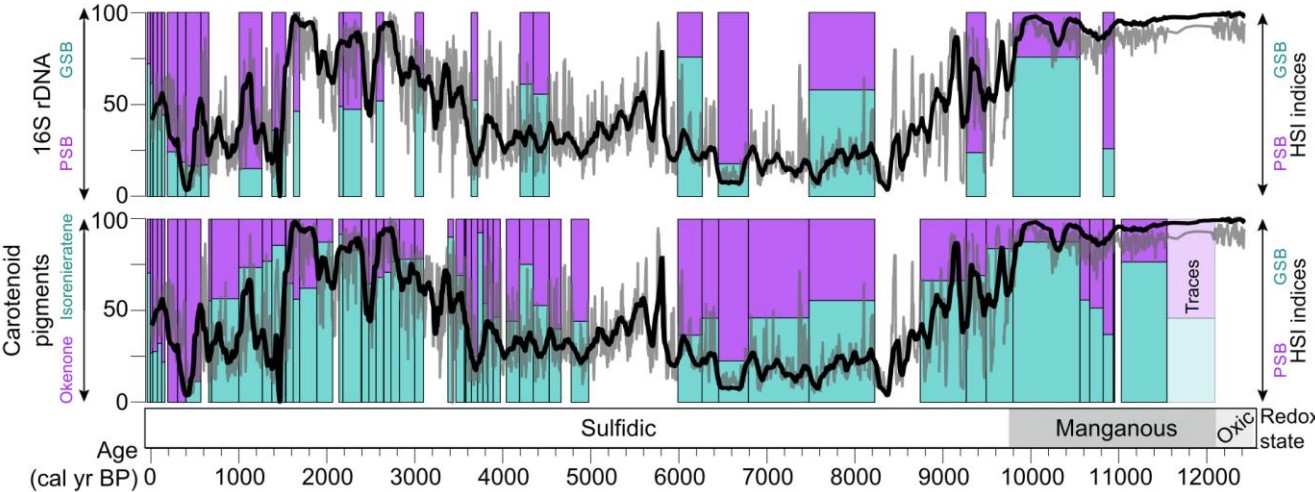

**Figure 4: Comparison of the HSI method with previously published biomarker and DNA measurements of the relative proportions of PSB and GSB (Wirth et al., 2013; Ravasi et al., 2012). The relative ratio of the HSI indices is plotted as a 100-year running mean (black line) and 5-year running mean (gray line) in both panels.**

The close similarity of the spectral profiles of Chl-*a* and BChl-*c*, -*d*, and -*e* (and derivatives) when dissolved in organic solvent, makes it difficult to measure these pigments in samples that contain a mixture of both groups (Taniguchi and Lindsey, 2021). In particular, formulas developed for spectrophotometer measurements of these pigments yield inaccurate results when applied





to such mixed samples (Picazo et al., 2011). However, the absorption maximum of these pigment groups differ when they are *in situ*, with BChl-*c*, -*d*, and -*e* shifted to 710-750 nm compared to the 660-675 nm absorption maxima of Chl-*a* (Oren, 2011). This makes *in situ* HSI uniquely valuable as an efficient method to measure relative abundances of GSB-related pigments in sediments, especially those that contain pigments of both oxygenic and anoxygenic phototrophs.

The absorbance trough observed in our dataset at 710-730 nm fits most closely the absorbance spectra of BChl-*e* (Fig 2; Oren, 2011; Overmann et al., 1992), however BChl-*d* is likely also present and has an absorbance maximum at 715-745 nm that is overlapping. BChl-*c* is likely not present in significant amounts because it is not synthesized by the main species present in the most recent decades (Halm et al., 2009), and its absorbance is greatest at 745-755 nm, which typically near a reflectance maximum in the spectral endmembers. BChl-*e* has already been identified as the main GSB pigment in Lake Cadagno's water column during recent decades (Habicht et al., 2011; Gregersen et al., 2009; Posth et al., 2017), as well as in the sediments (Naeher et al., 2016). The presence of isorenieratene throughout the sedimentary record (Fig. 4; Wirth et al., 2013) confirms that the GSB population has consistently included brown-colored strains of GSB throughout the lake history, and these strains of GSB tend to utilize BChl-*e* for photosynthesis (Overmann et al., 1992; Sinninghe Damsté and Schouten, 2006). HPLC data also suggests the presence of Bphe-*e* and homologues (Fig. S2), however, without a reference standard to compare to, this identification remains uncertain.

**4.2 Long-term variations of phototrophic communities at Lake Cadagno**

The downcore HSI-inferred pigments reveal a high-resolution record of variable aquatic phototrophic communities in Lake Cadagno during the past 12,500 years (Fig. 5; dataset - Zander et al., 2023) that is consistent with the lower-resolution results of previously published DNA and carotenoid pigment analyses (Fig. 4; Wirth et al., 2013; Ravasi et al., 2012). However, the high-resolution HSI data provides more precise information about the timing of changes, and allows for a more detailed analysis of potential factors that might have driven community shifts. In this section, we compare the records of pigments with proxy data from Lake Cadagno and the adjacent region to possibly identify the main drivers of variations in the phototrophic groups over the Holocene.

In the early part of the record (12.5-10.2 kyr BP), sediments are dominantly clastic, and only traces of pigments are detected in the HSI data, making interpretations difficult. However, the more sensitive pigment measurement method used by Wirth et al., (2013) suggests that both PSB and GSB populations were already present over this interval, with a dominance of GSB (Fig. 4). Recent investigations of Lake Cadagno sediment porewater chemistry suggest that oxic groundwater is currently intruding into the lower lake sediments (Berg et al., 2022). Thus, it may be possible that pigments have been more degraded in the this section compared to the rest of the profile, which has been continuously anoxic.



Figure 5: Summary of proxies from Lake Cadagno and the alpine region. A-C) HSI pigment records with data plotted as calibrated pigment concentrations for TChl and Bphe-*a* (this study). D-E) Flood frequency and Mo XRF data (Wirth et al., 2013). F) Summer temperature reconstruction for the Alps based on a compilation of chironomids records (Heiri et al., 2015). G) Phases of glacier advance at Steingletscher, Swiss Alps (Schimmelpfennig et al., 2022). H) Pollen records of vegetation from Lake Cadagno (Vescovi et al., 2018)



The redox transition from clastic, manganous sediments to sulfidic conditions at 10.5 to 9.8 kyr BP is particularly interesting to investigate at high-resolution (Fig. 6A). Around 10.2 kyr BP, at the transition point from clastic-dominated sediments to increasingly organic-rich sediments, TChl becomes consistently detectable. Mo begins to increase at the same time as TChl, suggesting that the increase in oxygenic photosynthesis may have contributed to the onset of sulfidic conditions through increased oxygen consumption in the bottom layers due to organic matter remineralization. Approximately 2 cm (~100 years)

above this transition, the GSB index ($RABD_{710-730max}$) increases substantially, indicating the onset of euxinic conditions within the lake. The GSB index appears to match the Mo profile at the redox boundary, suggesting a link between more sulfidic conditions and GSB development at this time. Another ~150 years later, PSB pigments appear for the first time, most likely indicating a shallowing of the chemocline based on the higher light intensities required by PSB compared to GSB. These changes occurred during a time of relatively warm temperatures (Heiri et al., 2015; Fig. 5F) and forests were already

established in the catchment for approximately 2000 years (Vescovi et al., 2018; Fig. 5H). Early Holocene warming and drier summers would have contributed to permafrost melting, soil stabilization, and more solute-rich groundwater flows to the lake, which led to the development of crenogenic meromixis. Decreasing frequency of flooding also enabled meromictic conditions.

Wirth et al., (2013) concluded that Lake Cadagno was continuously meromictic with a sulfidic monimolimnion during the past

~10 kyr, and this result is confirmed by our high-resolution record of bacterial pigments (Fig. 5). Nonetheless, the pigment records show substantial shifts in the phototrophic communities during this period. From 10.2-8.8 kyr BP all three pigment groups have high concentrations indicating productive communities of PSB, GSB and oxic photosynthesizing algae. TChl and the GSB index reach maximal levels during this period. This period was likely defined by stable stratification with highly sulfidic conditions at the chemocline. Relatively high productivity in the mixolimnion may have shaded the chemocline enough

to favor GSB dominance over PSB. The next phase, from 8.8 to 3.4 kyr BP is dominated by PSB. GSB remain present throughout most of this interval, however at much lower concentrations than the previous phase. TChl decreases slightly compared to the previous phase. Decreased oxygenic productivity, likely driven by nutrient limitations, may have increased light availability at the chemocline enough to favor PSB over GSB. GSB production could have continued in extremely low-light conditions (due to dense PSB populations), but growth rates are substantially slowed under such conditions (Overmann

et al., 1992), leading to lower values of our GSB index. At approximately 5.8 kyr BP, the GSB index increases slightly. The timing of this shift matches with changes in several other proxies from Cadagano: tree pollen decreases due to Neolithic human disturbance (Vescovi et al., 2018; Fig. 5H), Mo counts decrease to a lower level, and the frequency of flood deposits increases. Together, these changes suggest increased mixing of the lake due to greater wind shear on the lake and more frequent flooding, which apparently led to more favorable conditions for GSB. Several studies have documented an increase in lake mixing due

to early-human deforestation and a consequent decrease in PSB; this dynamic occurs often in small deep lakes (Makri et al., 2020; Tu et al., 2021).



At ~3.4 kyr BP, Bphe-*a* concentrations drop to near zero, remaining at low values with occasional short-duration spikes until 1.3 kyr BP. During this time interval, the GSB index reaches higher levels, but is highly variable. TChl also increases, indicating an increase in algal productivity. The shift from PSB dominance to GSB dominance implies less light penetration at the chemocline either due to chemocline lowering or increased turbidity above the chemocline. This shift occurred during a period of high flood frequencies, and a peak in flood frequency at 2.5 kyr BP is associated with low levels of both PSB and GSB, suggesting frequent mixing and oxygenation of the water column by floods. However, equally high flood frequencies occurred around 3.9 kyr BP, when PSB were still abundant, suggesting that additional factors may be needed to explain the very low Bphe-*a* from 3.4-1.3 kyr BP. One hypothesis may be that climate changes (mainly cooler summers) following the onset of the Neoglacial period in the Alps led to a less stable stratification within the lake. Summer temperature reconstructions for the Alps from chironomids show a drop in temperature at approximately the same time as the sudden decline in PSB (Fig. 5F; Heiri et al., 2015). Additionally, glacial advances are recorded around 3-3.5 kyr BP at several sites in the Alps, such as the Steingletscher (Fig. 5G), Rhone Glacier and Grindelwald Glacier (Schimmelpfennig et al., 2022). Cooler summers would have weakened thermal stratification making the chemocline more susceptible to disturbance. Furthermore, groundwater input to the lake may have become more dilute in solute concentrations due to increased flow in the subsurface under neoglacial conditions, which may have also driven weaker stratification and lower sulfide availability. This could occur due to wetter conditions or increased snow cover, which would increase groundwater flow. Most likely, the severe decline in PSB from 3.4 to 1.3 kyr BP was driven by a combination of factors, including cooler temperatures, deforestation, and increased flood frequency, all of which increased water column disturbance in the lake and led to less light penetration at the chemocline.

From 600-1600 CE, during a period of relative warmth (Medieval Climate Anomaly; Büntgen et al., 2011), PSB returned to high levels, while GSB remained high and variable. This change aligns with an increase in Mo and a decline in flood frequency (Fig. 5), suggesting that the decrease in flood events led to a more stable water column and a shallowing of the chemocline (expansion of the sulfidic layer). During the past 400 years, concentrations of pigments decreased substantially, but evidence of all three phototrophic groups remain, with a slightly increasing portion of GSB. These changes could be due to increased flood frequency and summer cooling associated with the Little Ice Age (Büntgen et al., 2011).

To summarize, PSB and GSB communities are strongly affected by processes that impact their ecological requirements, i.e. anoxic/sulfidic conditions with light availability. PSB are more sensitive to changes in light availability and, therefore, were highly sensitive to processes that introduced turbidity or strengthened mixing in the mixolimnion, and thereby weakened the chemical gradients (i.e. $H_2S$ and $O_2$) at the chemocline. Correlation analyses shows a negative correlation between flood frequency and all three pigment groups, with the relationship being particularly strong for Bphe-*a* (r = -0.40; Fig. S3). Mo is positively correlated with all three groups, suggesting a link between sulfidic conditions and primary production. Hydrological conditions (floods and groundwater), land cover, and climatic conditions appear to have been key drivers of millennial scale



shifts in the composition of anoxygenic phototrophs community, likely causing cascading changes in biogeochemical cycles and ecosystem function (Bell et al., 2005).

**Figure 6: Detailed views of A) the redox transition interval 9.8 k yr BP, B) an interval with frequent flood deposits (2.8-2.6 k yr BP),**
380 **C) top of a mass movement deposit (MMD 5.9 k yr BP), and D) second example of a MMD (9.1 k yr BP). Shown are core pictures with flood deposits marked in dark blue and MMD marked in orange, Mo concentrations at 1 mm down-core resolution (black), Bphe-*a* concentrations with spatial map (violet), TChl concentrations with spatial map (green), and RABD₇₁₀-₇₃₀max index (GSB index; light blue). The red vertical lines on the hyperspectral images mark the location of the plotted downcore profiles. Yellow dashed lines indicate tops of MMD. Gray rectangle in panel A shows redox transition zone. Pink rectangles in C and D highlight**
385 **lags between reestablishment of GSB and PSB following event deposits.**



### 4.3 Short-term response of phototrophic communities to floods and mass movements

High-resolution HSI pigment data enables a detailed investigation of the changes in the phototrophic communities to short-lived events such as floods and mass movements that likely disturbed the ecological conditions required by the anoxygenic phototrophic bacteria. Floods represent disturbance events that cause short-term changes in ecological conditions within the lake by injecting oxygen into the monimolimnion through undercurrents (Fink et al., 2016), by decreasing light penetration via delivery of suspended sediment in interflows and overflows, or mixing the water column (Sabatier et al., 2022). Mass movements of previously deposited sediment may also decrease light penetration, mixing the water column or otherwise alter biogeochemical conditions in the water column. In this section, we analyze several example sections in detail to describe the changes in the phototrophic communities in response to floods and mass movements.

Frequent, but relatively thin (1 to 3 mm), flood deposits are found in the sedimentary record at 2.8 to 2.6 kyr BP (Fig. 6B) making this interval well suited for studying the recovery of the different phototrophic groups in between small flood events. The intervals from 405 to 372 mm and from 348 to 320 mm section depth (Fig. 6B) are characterized by flood deposits intercalated in background sediments, where 1 mm of background sediment corresponds to ~3 years. Here, TChl reaches high values in the regular sediments in between flood deposits. No evidence of PSB is found in the HSI data, while GSB pigments are found in distinct layers. During the interval from 372 to 348 mm, virtually no background sediment is present in between flood deposits and consequently all pigment indices are at low levels. The lack of PSB in this interval of frequent flooding suggests that there was persistently less light radiation at the chemocline either due to a lowering of the chemocline, or increased turbidity above the chemocline. GSB are more tolerant of low-light conditions (Van Gemerden and Mas, 1995) and, therefore, are present during this interval, however, not continuously. Discrete layers of regular lacustrine sediments (indicated by high TChl) that do not show evidence of anoxygenic phototrophs might represent periods of greater lake mixing when oxygenation of the water column limited growth of PSB and GSB. The response of the phototrophic communities to floods is, however, variable. In other parts of the record with less frequent flooding, there are instances with PSB and GSB pigments return immediately when normal lake sedimentation resumes, indicating no major impact of the flood event on these communities. The variable response of the bacterial communities to floods is likely controlled by the magnitude of the flood as well as how the flood interacts with the lake water, i.e., via underflows and turbidity currents, or interflows, which would result in suspended sediment and oxygen being delivered differently to the water column.

Examples of mass movement deposits (MMDs) are shown in Fig. 6C and 6D; both are slumps composed mainly of previously deposited lake sediment (note pigment-rich sediment within the MMD). At 5.9 kyr BP (Fig. 6C) a mass movement deposit is overlain by 7 mm in place lacustrine sediments that first contain mainly low concentrations of TChl and a lack of PSB and GSB. This suggests that the MMD substantially disturbed the water column leading to an extended period (approximately 60 years based on the uncertain sedimentation rate in this section) of oxic conditions through the lake. High concentrations of




manganese at the upper boundary of MMDs confirm that oxygen was introduced to the sediment water interface during these
events (Wirth et al., 2013). GSB and PSB pigments both increase sharply after this period, but GSB lead the PSB pigments by
about 2.5 mm (~20 years), suggesting a delay in the reestablishment of PSB following the MMD, most likely due to a deeper
chemocline with too little light penetration for PSB growth. The sharp increase in GSB likely represents a transition from
seasonally oxic conditions to permanently anoxic conditions in the hypolimnion, whereas the sharp increase in PSB likely
represents the time when the chemocline shallowed enough for there to be sufficient light to produce blooms of PSB. A similar
pattern is observed for an older MMD illustrated in Fig. 6D. In this case, GSB production resumed immediately following the
MMD, though at somewhat lower levels, perhaps due to a light limitation. After about 60 years (4.5 mm), PSB return to high
levels, and GSB production also increases substantially, indicating a return to meromictic conditions with a stable and shallow
chemocline.

These case studies of selected intervals show that floods and mass movement events within the lake can cause significant
disturbance to the anoxygenic phototrophic communities within the lake, with effects lasting several decades, though these
temporal estimates are uncertain due to low sedimentation rates. Nonetheless, the observed delay between the re-establishment
of PSB and GSB appears in several instances, and is robustly measured by the high-resolution HSI data. Such information on
mm-scale could not be observed through conventional pigment measurements. These findings are interesting to compare with
a recent drastic change in the bacterial community from a dominance of PSB to a dominance of GSB in 2000 AD that was
triggered by water-column mixing due to a storm (Tonolla et al., 2005). Following the storm, GSB biomass increased
approximately threefold, while PSB biomass remained stable (Tonolla et al., 2017). One potential explanation for this shift
could be increased availability of nutrients, carbon and/or sulfide at the chemocline due to the mixing event (Decristophoris et
al., 2009). As presented above, similar processes may have been associated with past flooding or mass movements that
disturbed the water column and favored GSB over PSB.

## 5. Conclusions and outlook

Our results demonstrate that HSI core scanning is an effective technique for tracking the relative abundance of three different
pigment groups: TChl (a proxy for oxygenic phototrophs), Bphe-*a* (a proxy for purple sulfur bacteria), and BChl-*c*, -*d*, and -*e*
(a proxy for green sulfur bacteria). Our study is the first, to our knowledge, to show that HSI core scanning can be used to
detect GSB-related bacteriochlorophylls, and we show that the method works even in sediments containing pigments from
oxygenic and anoxygenic phototrophs. The reflectance spectra of TChl and BChl-c, -*d*, and -*e* are distinct using this *in situ*
method, offering an advantage over spectrophotometric methods on pigments dissolved in solvents in which the spectra of
these two pigment groups overlap and are essentially indistinguishable in mixed samples. The HSI pigment records confirm
previous work showing nearly continuous meromictic conditions throughout the Holocene and substantial shifts in anoxygenic
phototrophic bacteria communities (Wirth et al., 2013). The high-resolution data from the HSI method make it possible to



investigate how disturbance events such as floods and mass movements might have impacted phototrophic communities with unprecedented detail. We suggest that flood and mass movement events repeatedly disturbed the lake ecosystem during the Holocene and possibly disrupted the biogeochemical conditions required by anoxygenic phototrophic bacteria over seasonal to decadal timescales. Also, the inflow of groundwater into the lake and the solute concentration of these waters likely varied

in the course of the Holocene. This certainly had an impact on the lake water properties and, therefore, on the phototrophic communities. Our results show that PSB are more sensitive to disturbance events, and their recovery to previous levels can take several decades longer than that of GSB and oxygenic phototrophs. Through comparison with other proxy records, we conclude that allogenic environmental factors, such as floods, air temperature, and land cover, drove major shifts in the structure of the Lake Cadagno phototrophic communities.


Future work may refine this initial work to validate the use of *in-situ* sediment reflectance for detecting GSB-related BChl. A quantitative calibration of our GSB index was not possible in the scope of this work, but would be possible via a more detailed investigation of the sedimentary pigment assemblage to quantify total BChl-*c*, -*d*, -*e*. It may also be possible to improve the accuracy of HSI estimates of pigments in sediments with multiple pigment groups by using multivariate regression techniques

or spectral deconvolution to isolate the signal of each pigment group.

*Author contributions.* PDZ: Conceptualization, Formal analysis, Investigation, Writing - original draft, Visualization, Writing – Reviewing and Editing.  SBW: Conceptualization, Formal analysis, Investigation, Writing - original draft, Visualization, Writing – Reviewing and Editing. AG: Investigation, Writing – Reviewing and Editing. SP: Writing – Reviewing and Editing.

MG: Conceptualization, Writing - Review & Editing, Resources, Supervision.

*Code and data availability.* Data generated for this manuscript (hyperspectral imaging indices and pigment measurements) are available at: https://doi.org/10.5281/zenodo.7573508.

*Competing interests*. The authors declare no competing interests.

*Acknowledgements.* This work was funded by Swiss National Science Foundation (SNF) grants 200021_172586, P500PN_206731, and 200021-121909. We thank the Alpine Biology Centre Foundation for logistic support and coordination during fieldwork and sampling on Lake Cadagno. Andrea Sanchini, Patrick Neuhaus, Daniela Fischer and René Nussbaumer

assisted with HPLC pigment measurements.



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
