# Peer review of "Hyperspectral imaging sediment core scanning tracks high-resolution Holocene variations in (an)oxygenic phototrophic communities at Lake Cadagno, Swiss Alps"

_EGUsphere, 2023_

## Author Comment (AC1)

**Response to Anonymous Referee #1**

We thank the reviewer for the constructive review. Our responses to their comments are in red below.

The paper is well-written and the idea is ueful for the community. however, there are some doubts about reliability of the new proposed index which require more debates statistically. Here are my comments:

RABD670: There is a review from Van Exem et al. 2022 which they reviewed all the chl indices and they found out that RABAs work better than RABDs. Have you tried to implement that and compare the results since it seems to be possible with your data.

We did test RABA (relative absorption band area) indices, and found that the correlations between RABA indices and pigment measurements were slightly lower than the RABD indices we selected. We expect that the RABA indices are more affected by the overlapping absorption of chlorophyll-*a* pigments and Bchl-*c, -d, -e* pigments, and indeed we see that $RABA_{640-695}$ is significantly correlated with isorenieratene, suggesting an influence of overlapping pigments from GSB. For these reasons, we chose to proceed with the selected RABD indices. The comparison of RABA vs RABD indices was already shown in Fig. S1. We will include the correlation values used to generate this figure in a table in the supplement so that readers can see the precise values. We also note that the Van Exem et al. study presents results from one site and it is not clear if RABA indices work better than RABD indices at all sites.

Line 160: how did you determine the percentage reflectance values to remove cracks and mineral reflections? it seems it is based on practical excercise on your core and your hyperspectral scanner which it should be noted in the text.

Yes, this was done by some experimentation to achieve satisfactory results based on the data from our cores. These thresholds might not be relevant for other cores or scanners.

Fig 2: since RABD710-730 is located in the right shoulder of chl-a and phaephytin-a absorption bands, how would they effect this index? with bigger amount of chl-a and Phae-a it is expected that GSB-related signal disapear. And based on your data was there any detection limitation on this index?

Yes, there can be an influence of Chl-*a* and pheophytin-*a* on the $RABD_{710-730max}$ index. We have attempted to limit this influence by defining the trough to be from 695-765 nm, which, by having the start point on the slope of the absorption of Chl-*a,* limits the influence of Chl-*a* on the $RABD_{710-730max}$ value. Using this formula, there is likely to be a slight bias towards lower $RABD_{710-730max}$ values when Chl-*a* concentrations are high. We considered other formulations with wider trough endpoints (640-765 nm), however this version of the index is more strongly correlated with RABD670, indicating a greater influence of Chl-*a,* where the $RABD_{710-730max}$ is increased by high amounts of Chl-*a*. We did not attempt to formally calculate a detection limit because of the challenge of assessing all possible pigments that could be contributing to this index. We suggest that a greater number of pigment samples should be processed, and a more thorough determination of GSB pigment compounds would also be necessary to determine a robust detection limit. However, a comment below stimulated a rough calculation of a detection limit, see below.

Fig3: It would be better to remake all the plots in a way that the whole dots and the range are observed.

We will implement the suggested change.

Fig 3C: There are some RABD710-730 dots which are valued less than one, how do you interpret them?

RABD values less than 1 indicate that spectral profile is convex over the interval specified by the index. This is controlled by the reflective properties of the sediment. While RABD = 1 has been interpreted as generally indicating no absorption of the substance of interest, if the sediment has a convex spectral profile in the absence of any pigment, then a small amount of pigment could yield an RABD value of less than one. As a thought experiment, imagine that a sediment with no pigments has an RABD of 0.95, then adding some detectable amount of pigment to the same sediment might produce an RABD of 0.98. Therefore, we don't interpret RABD values less than 1 any differently than other RABD values. In the example of RABD710-730, a convex shape is likely to occur over the interval defined by the index in part due to the shape of the right side of the absorbance trough of Chl-*a*, which does have a convex shape. Additionally, several mineral components can also have a convex reflectance profiles over this interval (Kokaly et al., 2017).

Moreover, considering RABD710-730, I am curios to see what correlation you will get if you remove Isorenieratene which have big values ( over 1000) and then recalculate the correaltion. It seems the correlation for the samples with Iso under 1000 are weak. Maybe, it can be discussed in terms of limitations of the index!

Regarding the correlation of isorenieratene and RABD710-730 – removing all data points with isorenieratene measurements greater than 1000 area/g yields an insignificant correlation, however, this is based on only 8 data points. We find that the correlation remains significant at $p < 0.05$ if we remove the 4 greatest values (>3000 area/g), leaving n = 17 samples.  If we remove more samples, the correlation is weakened and no longer significant. This suggests a conservative detection limit around RABD710-730 = 1.036 (the highest RABD value of the lowest 17 samples), though a greater number of samples might clarify this point. We will mention this in the text to indicate that there is greater uncertainty at lower values, and that variations in the RABD index below about 1.036 should be interpreted cautiously, or considered below the limit of detection.

Fig4: this figure is a bit unclear. suggest to change the caption and specify y-axes is  related to which RABD.

We agree with the suggestion, and will modify the figure and caption to make it more clear which data corresponds to which axis.

Finally, it is always a question that a model or here an index which is applied on one core can be applied on any other cores? what would be the limitations and maybe a discussion on this in the paper would be useful for reader.

We expect that newly developed RABD710-730 index can be applied to other sediments with high amounts of GSB-pigments. Ongoing unpublished work on Soppensee and other lakes confirms these initial results. Most likely the biggest limitation is the concentration of GSB-related bacteriopheopigments; it may be relatively rare to reach high enough concentrations in sediments for the HSI method to be able to robustly detect these pigments. We will include this in the revised discussion.

**References**

Kokaly, R.F., Clark, R.N., Swayze, G.A., Livo, K.E., Hoefen, T.M., Pearson, N.C., Wise, R.A., Benzel, W.M., Lowers, H.A., Driscoll, R.L., and Klein, A.J., 2017, USGS Spectral Library Version 7: U.S. Geological Survey Data Series 1035, 61 p., https://doi.org/10.3133/ds1035.

---

## Author Comment (AC2)

**Response to Anonymous Referee #2**

We thank the reviewer for their constructive review. Our responses to their comments are in red below.

**Overview**

Zander et al. use recently developed hyperspectral imaging methods, including one first introduced in this manuscript, to quantify the abundance of sedimentary photopigment produced by oxygenic phototrophs and anoxygenic phototrophs - specifically purple and green sulfur bacteria (PSB and GSB, respectively). They apply this technique to Lake Cadagno, a lake that has been well studied for both modern and paleolimnology. The authors find that total chloropigments (TChl) and purple sulfur bacteria (PSB) are readily detectable using key absorption troughs in hyperspectal imaging, and calibrate their abundance using concentrations measured on samples using extraction and spectrophotometry and HPLC. They argue that GSB are detectable by a characteristic absorption trough, and around 700 nm, however it is difficult to calibrate as the pigments are not readily measured by other techniques. Instead they use co produced carotenoids to validate the detection of GSB, but do not attempt a calibration, instead they interpret the index as an indicator of relative abundance.

The detection and interpretation of these pigments by hyperspectral imaging has great potential for paleolimnological studies. HSI methods are rapid, high-resolution and non destructive, and analyzing these pigments provides important and direct inference on the past light and oxygenation structure of the lake. GSB is also quite difficult to measure, and it's possible that in situ detection via HSI may actually work better than other approaches, although this is difficult to prove. In the case of Lake Cadagno, the pigment records provide important insights on both long-term lake evolution, and response to external forcings and pressures, but also the response of the lake to flood events, highlighting the value of the high resolution detection.

The manuscript is well written and well illustrated, and effectively articulates the importance and value of the approach. It's an important contribution to the paleolimnologic literature, and I recommend it for publication following the correction of a minor concerns.

**Minor concerns:**

Non-normal distributions of pigment and RABD data

My only substantial concern with analysis and the results has to do with the treatment of the data in the calibration and reconstruction approaches. Specifically, I worry about non-normality in both the the pigment concentrations measured by spectroscopy and HPLC, and in the RABD indices. Both datasets are left bounded, and tend to be skewed right. Some strongly so. On line 236 the authors note that this may hamper their interpretations. The authors use linear regression to relate RABD indices to TChla and PSB, and to quantify their uncertainties, but this approach assumes normality in both the predictor and predictand. Looking at figure 3, this seems reasonable for TChl-a and RABD670, but the PSB data look right skewed, and potentially bimodal. And looking at figure 5, all the downcore HSI data, and especially the GSB index and PSB reconstruction both look left bounded and right skewed, as expected. I'd

encourage the authors to either demonstrate normality for these data, or consider transforming them before calibration, or use a different approach that does not require normally-distributed data.

I believe this is a common misconception. Normality of X and Y variables is not required for linear regression, the only requirement for normality is in the residuals of the regression models (Poole and O'Farrell, 1971)

Shapiro-Wilk normality tests on the residuals of both regression models used in this study yield $p$-values > 0.05, suggesting there is not significant evidence to suggest that the model residuals are non-normal. We will include a plot of the residuals and their distribution in the revised supplement.

However, the reviewer is correct in that non-normality may be a problem for the significance test we used for the correlation, i.e. the $p$-values reported in Fig. 3. We have now calculated the correlations using a non-parametric permutation test, which does not require an assumption about normality of the variables. Furthermore, we adjusted the $p$-values for the false discovery rate of multiple hypothesis testing using the Benjamini-Hochberg procedure (Benjamini and Hochberg, 1995). The results of the permutation test using 10,000 permutations show that the correlations remain significant for all relationships shown in Fig. 3. We will update the methods section to describe the updated statistical methods.

For the Bphe-a and RABD842, I wonder whether the data really support a continuous calibration, or whether a on/off, presence/absence type calibration would be more appropriate. To my eye (Fig 3A), there are basically two populations of Bphe-a concentrations. Those close to zero, and those around 1000 ug/g. The lower concentration population does suggest a relationship, but the trend is notably lower than that inferred from the dataset as a whole. The two highest Bphe-a concentrations do also stand out, but it's hard to know how much to weight those outliers. I'm open to the argument for a continuous relationship, but I think it needs to be made as it seems plausible that we're mostly seeing presence/absence of PSB in these data.

The reviewer is correct in pointing out that there is some evidence for a discontinuity in the linear relationship between Bphe-*a* concentrations and RABD842. However, I would not agree that it is a presence/absence type of calibration because there is a considerable range of concentrations in the lower population (0-400 ug/g), which appears to follow an approximately linear relationship based on a limited sample size, although it is true that a linear model fit to only this population would have a lower slope than the general model.

It is possible to consider that a non-linear model could be used to fit this relationship better, and indeed a quadratic polynomial does improve the model fit ($R^2$ = 0.938 vs 0.920). Exponential and power models yield a poorer fit. Given that the improvement of the quadratic model is rather small, we chose to use a linear calibration model based on the reasoning that linear calibration models have previously been the dominant choice for calibrating sediment reflectance absorption trough indices and pigment concentrations (Rein and Sirocko, 2002; Wolfe et al., 2006; Butz et al., 2015; Zander et al., 2022).

It is difficult to explain the apparent jump in pigment concentrations observed in the scatterplot at RABD values around 1.3-1.4, but given measurement uncertainties, limited number of samples, and other sources of error (e.g. discrepancies between pigment abundances at the

core surface compared to the underlying sampled sediments), I would not overinterpret this feature.

**Other Concerns**

Figure 4.

The comparison between the aeDNA data and the PSB and GSB indices is very interesting. I like the presentation in figure 4, but would be very keen to see a direct comparison of the aeDNA sample data and the HSI-based ratio. Could you add a scatter plot where the PSB/GSB indices over the DNA sample depth ranges are averaged and the plotted against a similar metric in the DNA data? There are intervals where it seems to agree quite well, but others where it doesn't, and I'd like to see the comparison explicitly, while also accounting for the different resolution of the datasets.

Based on the reviewer's suggestion, I have made this plot. The relationship between the relative HSI index for BSG vs PSB and the 2013 carotenoid pigment data is statistically significant, but with a low $R^2$. Comparing the relative HSI index with the DNA data, the relationship is not statistically significant. This comparison is challenging because the samples for carotenoids and DNA in the 2013 study were taken from different cores, not the same ones as the HSI. The correlation of the sample intervals to the hyperspectral imaging data may include some mismatches, in particular in sections with many floods and slumps, which can have different thicknesses in the two cores. Furthermore, several of the 2013 samples include substantial material from mass movement deposits (slumps and floods). The mass movement deposits tend to have much lower concentrations of pigments, making the data from HSI in these sections noisy and mostly unreliable. Additionally, the 2013 carotenoid data and DNA are reported only as relative percentages of GSB vs PSB material. It is not trivial to relate that type of data to the hyperspectral indices, which are related to concentration, but have different relative scales due to the different absorption of the pigments. We attempted to account for this by standardizing the RABD indices to a 0 to 1 scale, and then taking their ratio. Altogether, these challenges make it not possible to compare the two datasets in an "apples to apples" manner, and therefore, it is difficult to interpret the results of these scatterplots, and we do not plan to include them in the manuscript.

[Figure]

[Figure]

As a small note, in Figure 4 the HSI data shows only data from regular background lacustrine sediment, all mass movement deposits have been removed, whereas in the above scatterplots, the mass movement deposits were not removed if they were also included in the carotenoid/DNA samples from the 2013 paper. We will add some text to explain the possible reasons for discrepancies between our data and the 2013 data.

Figure 5. Add a Tchl label to panel A to match panels B and C.

The label will be added

Figure 6. Continuing the Tchl, PSB and GSB labeling on the curves would benefit figure 6 too.

The labels will be added

**References**

Benjamini, Y. and Hochberg, Y.: Controlling the False Discovery Rate: A Practical and Powerful Approach to Multiple Testing, Journal of the Royal Statistical Society: Series B (Methodological), 57, 289–300, https://doi.org/10.1111/j.2517-6161.1995.tb02031.x, 1995.

Butz, C., Grosjean, M., Fischer, D., Wunderle, S., Tylmann, W., and Rein, B.: Hyperspectral imaging spectroscopy: a promising method for the biogeochemical analysis of lake sediments, Journal of Applied Remote Sensing, 9, 096031, https://doi.org/10.1117/1.jrs.9.096031, 2015.

Poole, M. A. and O'Farrell, P. N.: The Assumptions of the Linear Regression Model, Transactions of the Institute of British Geographers, 145–158, https://doi.org/10.2307/621706, 1971.

Rein, B. and Sirocko, F.: In-situ reflectance spectroscopy - Analysing techniques for high-resolution pigment logging in sediment cores, International Journal of Earth Sciences, 91, 950–954, https://doi.org/10.1007/s00531-002-0264-0, 2002.

Wolfe, A. P., Vinebrooke, R. D., Michelutti, N., Rivard, B., and Das, B.: Experimental calibration of lake-sediment spectral reflectance to chlorophyll a concentrations: Methodology and paleolimnological validation, Journal of Paleolimnology, https://doi.org/10.1007/s10933-006-0006-6, 2006.

Zander, P. D., Wienhues, G., and Grosjean, M.: Scanning Hyperspectral Imaging for In Situ Biogeochemical Analysis of Lake Sediment Cores: Review of Recent Developments, Journal of Imaging, 8, 58, https://doi.org/10.3390/jimaging8030058, 2022.